# May the FORCE be with you?

A Framework for ODRL Rule Compliance through Evaluation

Wout Slabbinck[1,*], Julián Rojas Meléndez[1], Beatriz Esteves[1], Ruben Verborgh[1] and Pieter Colpaert[1]

[1]*IDLab, Department of Electronics and Information Systems, Ghent University - imec, Belgium*

### Abstract

Driven by data protection laws, interest in decentralized data sharing solutions has surged in recent years. To implement these solutions effectively, usage control is essential for ensuring regulatory compliance and interoperability. The Open Digital Rights Language (ODRL) standard stands out as a potential unified language for formulating fine-grained usage control policies, which must be consistently enforced through a policy engine across every node in the decentralized network. However, how to reliably and uniformly interpret ODRL policies remains an open problem, often leaving implementers to devise their own solutions. To address this problem, this work introduces the Framework for ODRL Rule Compliance through Evaluation (FORCE), designed to assist in policy development and enhance comprehension of evaluation outputs. FORCE is built upon the ODRL Evaluator – a component rigorously tested using the ODRL Test Suite. FORCE's main contributions are *i)* a guide on the several specifications and pieces of software we are building to work with the current standard and to test new features for a possible future version of the standard, and *ii)* a Web application that acts as a playground to test these specifications and software libraries. Future work could expand FORCE to include more powerful ODRL Evaluators that can deal with conflict resolution or policy instantiation.

### Keywords

Policy, ODRL, Usage Control

## 1. Introduction

To address privacy concerns arising from the usage of personal data by evolving digital technologies, the General Data Protection Regulation (GDPR) [1] was introduced as a framework to balance innovation with governance. One of its Articles highlights data portability[1], which gives users the right to receive their data and request its transfer from one platform to another.

---

*NeXt-generation Data Governance workshop 2025, co-located with 21th SEMANTiCS, Vienna, Austria*

*Corresponding author.

✉ wout.slabbinck@ugent.be (W. Slabbinck); julianandres.rojasmelendez@ugent.be (J. R. Meléndez); beatriz.esteves@UGent.be (B. Esteves); ruben.verborgh@ugent.be (R. Verborgh); pieter.colpaert@ugent.be (P. Colpaert)

🌐 https://woutslabbinck.com/ (W. Slabbinck); https://julianrojas.org/ (J. R. Meléndez); https://w3id.org/people/besteves (B. Esteves); https://ruben.verborgh.org/ (R. Verborgh); https://pietercolpaert.be/ (P. Colpaert)

🔾 0000-0002-3287-7312 (W. Slabbinck); 0000-0002-6645-1264 (J. R. Meléndez); 0000-0003-0259-7560 (B. Esteves); 0000-0002-8596-222X (R. Verborgh); 0000-0001-6917-2167 (P. Colpaert)

[1]Art. 20 GDPR: https://gdpr-info.eu/art-20-gdpr/

Another one emphasizes the right of access[2], entailing the right for users to request access to their data and, in addition, information regarding how that data is used. Conforming to these articles calls for a language that is both interoperable and reliable, ensuring systems can interpret and enforce them uniformly. Addressing the need for interoperability, the Open Digital Rights Language (ODRL) [2] serves as a versatile standard for defining semantic-based usage control policies. Unfortunately, a lack of formalisation in the standard about how to evaluate such policies results in varying interpretations and inconsistencies in policy enforcement [3, 4, 5, 6, 7, 8, 9].

To address challenges in interpreting ODRL policies, Slabbinck *et al.* proposed a systemic approach focused on developing interoperable policy engines [3]. They did this by introducing the **Compliance Report Model**, an interoperable model for the output of ODRL Evaluations, alongside an ODRL test suite for verifying implementations of ODRL Evaluators. Additionally, they provided a reference implementation of an ODRL Evaluator.

This article presents FORCE, the Framework for ODRL Rule Compliance through Evaluation, a specification and an open-source Web editor built upon the aforementioned ODRL Evaluator. The former contains a guide on how to design ODRL policies, understand ODRL evaluation and provides references to other specifications, including one about proposals for ODRL 3.0. The latter provides a live editor for inputting ODRL policies, requests, and the state of the world and it runs entirely in the browser without requiring an additional server. Furthermore, it generates machine-readable compliance reports with detailed human-readable explanations to aid user understanding. Finally, users can also load and refine test cases from the test suite to deepen their comprehension.

The remainder of the article is organized as follows: Section 2 discusses prior work on ODRL evaluation. In Section 3, we introduce the Framework for ODRL Rule Compliance through Evaluation and outline its key components. Finally, Section 4 concludes the paper.

## 2. Related Work

The Open Digital Rights Language (ODRL) Information Model 2.2 [2] is a W3C Recommendation to express policies. This standardisation effort to reach the aforementioned recommendation explicitly excluded access control or enforcement mechanisms [10]. Research about evaluating ODRL policies often mentions a lack of evaluation semantics, thus resulting in the enforcement not being interoperable with each other [3, 4, 5, 6, 7, 8, 9]. As a reaction, a task force within the ODRL Community Group was formed to create the ODRL Formal Semantics specification[3] to formalise the behaviour of an ODRL Evaluator, its inputs and its outputs. Slabbinck *et al.* [3] contributed to this effort through introducing three contributions: The **Compliance Report Model**[4], an interoperable vocabulary which articulates compliance with ODRL policies and the State of the World (SotW); a **test suite**[5] for ensuring consistent policy evaluation across different implementations; and the **ODRL Evaluator**[6], an implementation that systematically

---

[2]Art. 15 GDPR: https://gdpr-info.eu/art-15-gdpr/
[3]ODRL Formal Semantics specification https://w3c.github.io/odrl/formal-semantics
[4]ODRL Compliance Report Model: https://w3id.org/force/compliance-report
[5]Test suite Repository: https://zenodo.org/records/14290518
[6]ODRL Evaluator: https://w3id.org/force/evaluator

evaluates policies and generates compliance reports.

## 2.1. ODRL extensions

In addition to the evaluation behaviour of ODRL 2.2, several research efforts aim to extend the recommendation with new features. Fornara and Colombetti propose incorporating temporal properties to model the lifecycles of permissions, obligations, and prohibitions, enabling compliance monitoring[11]. Akaichi *et al.* introduce variable constraints, resolved by introducing a deterministic dereferencing mechanism tied to external sources [5]. Cimmino and Fornara provide six distinct theoretical proposals, including a more generic proposal for variables in an ODRL Policy that can change over time [12]. Esteves *et al.* [13] are developing a policy instantiation specification[7] that builds on Slabbinck *et al.* work on the Compliance Report Model and the ODRL Evaluator [3] to create an instantiated agreement that fulfils both the requester's and the data holder's policies and specifies agreed conditions for data exchange. As such, there is a clear need for an experimentation platform to test and demonstrate both ODRL Evaluators and proposed extensions to the language.

## 3. FORCE

This section introduces the Framework for ODRL Rule Compliance through Evaluation (FORCE). The first subsection focuses on the FORCE specification, and the second discusses the FORCE web demonstrator used to explore ODRL evaluations and proposed extensions.

### 3.1. FORCE Specification

The FORCE specification[8] serves as a guide for the development and evaluation of ODRL policies. It elaborates how the behaviour of the ODRL Evaluator and how ODRL Evaluators in general should be tested for correct behaviour, complementing the work of Slabbinck *et al.* [3]. Examples are included that clarify how the ODRL Compliance Reports, the evaluation results, are produced and interpreted. Furthermore, the specification introduces a suite of specifications and tools under development, designed to support both the current ODRL standard and experimentation with ODRL 3.0 proposals.

### 3.2. FORCE Web Demonstrator

This section describes FORCE Web editor[9], a demonstrator developed to aid the understanding of ODRL policy evaluation. This is achieved through three core features: An intuitive user interface to edit and evaluate policies, comprehensive output explanations, and a selection of scenarios to facilitate the exploration of ODRL evaluation examples.

The first feature is its intuitive design, achieved by implementing it as a Web application for evaluating ODRL policies. The ODRL Evaluator operates with three essential inputs: (i) an ODRL

---

[7]Policy instantiation specification: https://w3id.org/force/policy-instantiation
[8]FORCE specification: https://w3id.org/force
[9]FORCE web demonstrator: https://w3id.org/force/demo

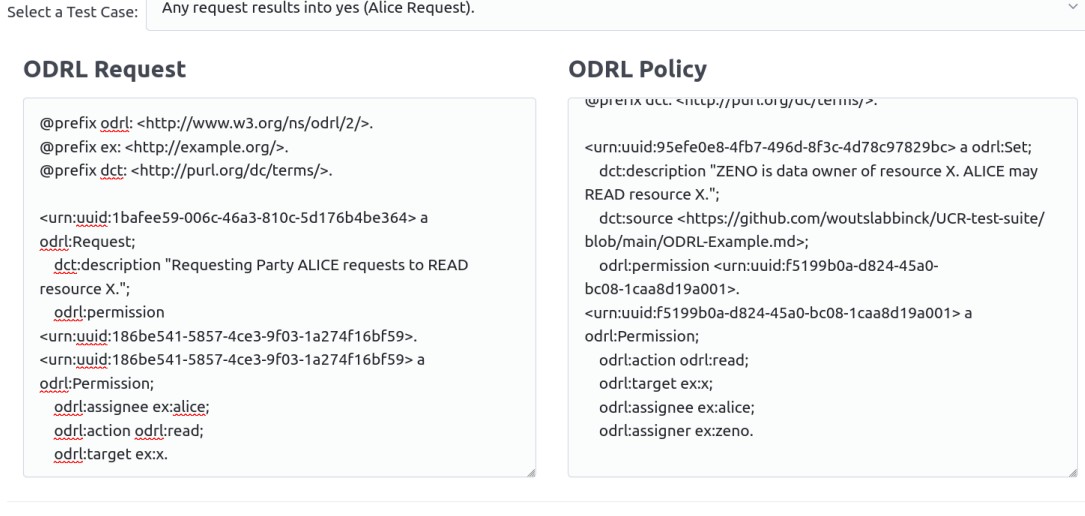

**Figure 1:** Screenshot of the FORCE Web editor displaying (i) a dropdown menu to select test cases from the test suite; (ii) three editable input fields (ODRL Request, ODRL Policy and State of the World); and (iii) an evaluate button, which initiates the ODRL evaluation resulting in the output appearing below the ODRL Compliance Report title.

policy, which outlines the rules defining what access is allowed to which resource by a specific entity; (ii) an ODRL request, containing an ODRL request policy that specifies the actions a party wishes to perform; and (iii) the state of the world, which provides additional context necessary for evaluating constraints. To accommodate these, the FORCE application offers three editable input fields. To further enhance the user experience, the application addresses the cold start problem by pre-loading an example, as illustrated in Figure 1. Evaluating over the input happens in the browser by relying on the EYE JS [14] library as the core reasoning engine of the ODRL Evaluator[10].

The second feature serves to reduce the intricacies of the ODRL Compliance Report Model. The machine-readable Compliance Report describes for each policy its associated rules, the activation state, and the rationale behind that state. To enhance clarity, the web editor includes a human-readable explanation highlighting the most important messages of the report. For example, if there is a `report:PermissionReport` that is `report:Active`, text is provided that entails that the requested action is allowed to be performed by the requesting party.

Furthermore, the demonstrator allows for showcasing a large number of handcrafted scenarios

---

[10]ODRL Evaluator npm package: https://www.npmjs.com/package/odrl-evaluator

as example inputs for ODRL evaluations. For that purpose, the test cases from the test suite introduced by Slabbinck *et al.* [3] are reused. A specific scenario can be selected through using the dropdown menu, as shown at the top of Figure 1. When a selection is made, the inputs are dynamically fetched thanks to the index hosted in the test-suite[11].

### 3.2.1. FORCE ODRL Extensions

To showcase how the FORCE Web interface can support ODRL extensions, we implemented the ODRL Dynamic Constraint extension introduced by Akaichi *et al.* [5]. Their approach resolves ambiguity in evaluating dynamic right operands in ODRL by introducing a new class (OperandReference) and a derefencing algorithm based on SHACL property paths[12]. An ODRL Constraint has two operands to which comparisons can be made with one relational operator:

- **Left Operand:** An instance of `odrl:LeftOperand`, linked via `odrl:leftOperand`. While ODRL defines its semantics, how to resolve its value remains unspecified[13].

- **Operator:** A comparison operator from the `odrl:Operator` class.

- **Right Operand:** Either a static value (`odrl:rightOperand`) or a dynamic reference (`odrl:rightOperandReference`) that must be dereferenced at evaluation time.

The challenge lies in comparing a left operand to a dynamic value. The proposed algorithm deterministically resolves the reference using SHACL paths and a dereferenceable URL. We've implemented this algorithm in the ODRL Evaluator and integrated several examples in the FORCE web demonstrator for ODRL extensions[14]. Further implementation details and a worked-out example are provided in the ODRL 3.0 proposals specification[15].

## 4. Conclusion

In this paper, we present FORCE as a set of specifications and a Web application that enhances user comprehension of ODRL policy evaluation. The Web editor enables users to create and edit ODRL policies, define requests, and configure various environment variables, all of which can be evaluated through an intuitive interface that also provides human-readable insights into compliance reports.

Through pointers to additional resources and proposals for ODRL 3.0, the FORCE specifications lay the groundwork for continued research and development. As an open-source initiative, FORCE facilitates extensibility and experimentation, enabling further research on ODRL evaluators. Future work includes addressing open challenges for ODRL, including conflict resolution, which addresses cases where multiple policies are incompatible, and policy instantiation, which involves defining the agreed-upon terms of exchange based on a set of policies and a request.

---

[11]Index of all the test cases of the test suite: https://github.com/SolidLabResearch/ODRL-Test-Suite/blob/main/data/index.ttl

[12]Property Path : https://www.w3.org/TR/shacl/#property-paths

[13]At the time of writing, no formal method exists to materialize left operand values.

[14]FORCE ODRL extension sandbox: https://w3id.org/force/demo/odrl3proposal

[15]FORCE ODRL 3.0 proposals: https://w3id.org/force/odrl3proposal

## Acknowledgments

This research was funded by SolidLab Vlaanderen (Flemish Government, EWI and RRF project VV023/10) and by the imec.icon project PACSOI (HBC.2023.0752), which was co-financed by imec and VLAIO and brings together the following partners: FAQIR Foundation, FAQIR Institute, MoveUP, Byteflies, AContrario, and Ghent University – IDLab.

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
