# OpenReview forum: "May the FORCE be with you? A Framework for ODRL Rule Compliance through Evaluation"
_SEMANTiCS.cc/2025/Workshop/NXDG — NXDG 2025_

### Official Review · ~Julio_Noe_Hernandez_Torres1 · 2025-07-18
**May the FORCE be with you? A Framework for ODRL Rule Compliance through Evaluation**

**Rating:** 6
**Confidence:** 4

**Review:**

## **Brief Summary of the Review**

This paper presents FORCE, a specification and a web demonstrator framework designed to support the evaluation of ODRL (Open Digital Rights Language) policies in decentralized data sharing contexts. The work addresses a critical gap in usage control and compliance verification by providing a human-readable and machine-readable evaluation interface, building upon existing efforts such as the ODRL Evaluator and Compliance Report Model.

The submission addresses a timely topic and makes a practical contribution to data governance frameworks that align with the GDPR. However, as a primarily systems/demo paper, it would benefit from stronger empirical evaluation, more explicit claims of originality, and a more structured comparison with related frameworks.

## **Review**


### **Strengths**
- **Relevance**: Addresses critical challenges in semantic usage control and next-generation data governance.
- **Practical Contribution**: Provides an open-source web-based tool (FORCE Web Demonstrator) aiding policy authors and developers.
- **Clarity and Accessibility**: Clear presentation with helpful diagrams and explanations.
- **Standards Contribution**: Actively contributes to ODRL 3.0 proposals and related W3C efforts.
- **User-Centric Design**: Web tool focuses on explainability and ease of use.

---

### **Weaknesses**
- **Originality**: The contribution is largely incremental over previous works (e.g., ODRL Evaluator, Compliance Report Model).
- **Lack of Evaluation**: No formal user study, performance assessment, or adoption analysis is presented.
- **Impact Analysis**: Limited exploration of how FORCE advances beyond current tools in a measurable way.
- **Related Work**: Comparison with similar frameworks (e.g., ODRE, Policy Reasoners) is minimal.
- **Scope**: Focused narrowly on ODRL without discussion on extensibility to broader policy models or usage control systems.

---

---

### Official Review · ~Renato_Iannella1 · 2025-07-22
**Review of ODRL FORCE paper**

**Rating:** 9
**Confidence:** 5

**Review:**

This paper discusses the evaluation of ODRL policies with the support of a "evaluation report model" standard.
The paper also shows the real use of the evaluation with an online demo.

The paper does present a good overview of the FORCE specification but does not explicitly show major parts of the language.
That maybe a good addition to make it clearer what is proposed.

Although this is just a suggestion, it would be useful to compare FORCE with the SHACL validation report: https://www.w3.org/TR/shacl/#validation-report

---

### Decision · Program_Chairs · 2025-07-25

Accept